# G-TRACER: Expected Sharpness Optimization

## Abstract

We propose a new regularization scheme for the optimization of deep learning architectures, G-TRACER ("Geometric TRACE Ratio"), which promotes generalization by seeking flat minima, and has a sound theoretical basis as an approximation to a natural-gradient descent based optimization of a generalized Bayes objective. By augmenting the loss function with a TRACER, curvature-regularized optimizers (eg SGD-TRACER and Adam-TRACER) are simple to implement as modifications to existing optimizers and don't require extensive tuning. We show that the method converges to a neighborhood (depending on the regularization strength) of a local minimum of the unregularized objective, and demonstrate competitive performance on a number of benchmark computer vision and NLP datasets, with a particular focus on challenging low signal-to-noise ratio problems.

## 1 Introduction

### 1.1 Problem setting

The connection between generalization performance and the loss-surface geometry of deep-learning architectures in the neighborhood of local minima has long been the subject of interest and speculation, dating back to the MDL-based arguments of (Hinton & van Camp, 1993) and (Hochreiter & Schmidhuber, 1997). The connection is an intuitively appealing one, in that the sharp local minima of the highly nonlinear, non-convex optimization problems associated with modern overparameterized deep learning architectures are more likely to be brittle and sensitive to perturbations in the parameters and training data, and thus lead to worse performance on unseen data. We can build some intuition for this from a probabilistic modelling perspective, given a dataset $\mathcal{D} = \{(x_i, y_i)_{i=1}^n\}$ consisting of $n$ independent input random variables $x_i$ with distribution $p(x)$ and corresponding targets (or labels) $y_i$ with distribution $p(y|x)$ and treating the parameters $w \in \Theta \subseteq \mathbb{R}^p$ of a deep neural network (DNN) $f(\cdot, w) : \mathbb{R}^{d_x} \to \mathbb{R}^{d_y}$ as a random variable. Given a loss function $l(y_i, f(x_i, w))$ our goal is to find a $w^*$ that minimizes the expected loss: $\mathbb{E}_{p(x,y)}[l(y, f(x, w))]$. Writing the finite-sample version of this expected loss as $L(w) = \sum_{i=1}^n l(y_i, f(x_i, w))$, we can form a generalized posterior distribution (Bissiri et al., 2016) $p(w|\mathcal{D}) = p(w)\frac{1}{Z}\exp\{-L(w))\}$ (with normalizer $Z$) over the weights, which coincides with the Bayesian posterior in the special case that the loss is the negative log-likelihood $L(w) = -\frac{1}{n}\sum_{i=1}^n \log p(y_i|x_i, w)$ and then, together with an output (conditional predictive) probability distribution $p(y|x, w)$, we can form the predictive distribution by marginalization:

$$p(y|x, \mathcal{D}) = \int p(y|x, w)p(w|\mathcal{D})dw \tag{1}$$

At a local maximum (or mode) $w_k$ of $p(w|\mathcal{D})$ we can can form the Laplace approximation (valid asymptotically, for large $n$):

$$p_k(w|\mathcal{D}) \approx \frac{1}{Z_k}p(w_k|\mathcal{D})\exp\left(-\frac{1}{2}(w - w_k)^T H(w - w_k)\right) \tag{2}$$

where (assuming, for simplicity, a flat prior) $H = \nabla_w^2 L|_{w=w_k}$ and the normalizer (which for the negative log-likelihood loss is the evidence, or marginal likelihood, and which we will also refer to as the *pseudo-marginal likelihood*) is given by $Z_k = p(w_k|\mathcal{D})(2\pi)^{\frac{d}{2}}\det(H)^{-\frac{1}{2}}$. Thus, in the neighborhood of each local maximum of $p(w|\mathcal{D})$, we approximate the posterior by a multivariate Gaussian with covariance given by the inverse

Hessian of the negative loss: $p(w_k|\mathcal{D}) \sim \mathcal{N}(w_k, H^{-1})$. Modern DNNs are characterized by multimodal losses (Wilson & Izmailov, 2020), and so, informally, we can decompose the posterior predictive distribution into disjoint contributions from each of the local maxima, the sum of which dominate the overall integral:

$$p(y|x, \mathcal{D}) \approx \frac{1}{Z} \sum_{k' \in \{k\}} \int p(y|x, w) Z_{k'} p_{k'}(w|\mathcal{D}) dw \tag{3}$$

where $Z = \sum_{k' \in \{k\}} Z_{k'}$, which is an expectation with respect to a probability measure with density given by: $\frac{1}{Z} \sum_{k' \in \{k\}} Z_{k'} p_{k'}(w|\mathcal{D})$, and which, by writing:

$$\frac{1}{Z} \sum_{k' \in \{k\}} Z_{k'} p_{k'}(w|\mathcal{D}) = \sum_{k' \in \{k\}} \pi_{k'} p_{k'}(w|\mathcal{D}) \tag{4}$$

can be viewed as a Gaussian mixture model (GMM) with mixing coefficients:

$$\pi_k = \frac{Z_k}{\sum_{k' \in \{k\}} Z_{k'}} \tag{5}$$

Thus the relative contribution of each component is given by the relative size of the pseudo-marginal likelihoods $Z_k$. For very high-dimensional $w \in \mathbb{R}^p$ (modern DNN architectures often have billions or even trillions of parameters) even small differences in the width of the Gaussian approximation will have exponentially large effects on the magnitude of $Z_k$ (which can be thought of as the the volume associated with the local maximum). How does all this relate to flatness? The Gaussian curvature $K$, providing an intrinsic (and thus coordinate-free) measure of curvature is given by:

$$K = \Pi_i \lambda_i = \det(H) \tag{6}$$

and contributions to the mixture thus scale inversely with $\sqrt{K}$. In other words, the flatter the solution, the more it contributes to the mixture model against which the output probability distribution is integrated, in order to form the posterior predictive distribution. In a typical high-dimensional setting, the effect of small differences in curvature (or flatness) is exponentially magnified. To see this, we can consider two local minima i) with Hessian $H$, and ii) an $\epsilon$-flattened minimum ($0 < \epsilon \ll 1$) with Hessian $H' = (1 - \epsilon)H$, (such that each eigenvalue of $H$ is simply shrunk by a constant factor $(1 - \epsilon)$). We have $K' = \det(H') = (1 - \epsilon)^p \det(H)$, so that the $\epsilon$-flattened minimum with curvature $K'$ has exponentially lower Gaussian curvature. The corresponding Gaussian approximations have covariances $\Sigma' \approx (1 + \epsilon)\Sigma$ and $\Sigma$, and the ratio of the corresponding pseudo-marginal likelihoods scales as:

$$\frac{\det((1 + \epsilon)\Sigma)}{\det(\Sigma)} = (1 + \epsilon)^p \xrightarrow{p \to \infty} \infty \tag{7}$$

Thus we can see that the contribution from the flatter minimum dominates in the high-dimensional limit. In an empirical study, Huang et al. (2019) trained a ResNet18 architecture on the Street View House Number (SVHN) dataset and estimate the volume around local minima using Monte-Carlo integration, finding that the volumes of basins surrounding minima that generalize well are at least 10,000 orders of magnitude larger than those of minima that generalize poorly.

A complementary approach is to characterize the loss-surface Hessian, since, at such local minimum of the loss, for a perturbation $\Delta w$, we have:

$$L(w + \Delta w) - L(w) = \Delta w^T \nabla^2 L(w) \Delta w + O(\|\Delta w\|^3) \tag{8}$$

There has therefore been a large literature attempting to characterize the loss-surface Hessian $\nabla^2 L(w)$ and to relate these characteristics to generalization. In many practically relevant cases, multiple minima are associated with zero (or close to zero) training error, and explicit or implicit regularization is needed to find solutions with the best generalization error. Overparameterization is associated with the bulk of the Hessian spectrum lying close to zero and thus to highly degenerate minima (Sagun et al., 2017). Wei & Schwab (2020) further show that given a degenerate valley in the loss surface, SGD on average decreases the trace of the Hessian, which is strongly suggestive of a connection between locally flat minima, overparameterization and generalization.

## 1.2 Sharpness-Aware Minimization

Despite the intuitive appeal and plausible justifications for flat solutions to be a goal of DNN optimization algorithms, there have been few practical unqualified successes in exploiting this connection to improve generalization performance. A notable exception is a recent algorithm, Sharpness Aware Minimization (SAM) (Foret et al., 2020), which seeks to improve generalization by optimizing a saddle-point problem of the form:

$$\min_{w} \max_{\|\Delta w\| \leq \rho} L(w + \Delta w) \tag{9}$$

An approximate solution to this problem is obtained by differentiating through the inner maximization, so that, given an approximate solution $\Delta w^* := \rho \frac{\nabla L(w^k)}{\|\nabla L(w^k)\|_2}$ to the inner maximization (dual norm) problem:

$$\arg \max_{\|\Delta w\| \leq \rho} L(w + \Delta w) \tag{10}$$

the gradient of the SAM objective is approximated as follows:

$$\nabla_w \left( \max_{\|\Delta w\|_{FR} \leq \epsilon} L(w + \Delta w) \right) \approx \nabla_w L(w + \Delta w^*) \approx \nabla_w L(w)|_{w + \Delta w^*} \tag{11}$$

While the method has gained widespread attention, and state-of-the-art performance has been demonstrated on several benchmark datasets, it remains relatively poorly understood, and the motivation and connection to sharpness is questionable given that the Euclidian norm-ball isn't invariant to changes in coordinates. Given a 1-1 mapping $g : \Theta' \to \Theta$ we can reparameterize our DNN $f(\cdot, w)$ using the "pullback" $g^*(f)(\cdot, \nu) := f(\cdot, g(\nu))$ under which, crucially, the underlying prediction function $f(\cdot, w) : \mathbb{R}^{d_x} \to \mathbb{R}^{d_y}$ (and therefore the loss) itself is invariant, since, for $\nu = g^{-1}(w)$, we have $f(\cdot, w) = f(\cdot, g(\nu))$. Under this coordinate transformation, however, the Hessian at a critical point transforms as (Dinh et al., 2017):

$$\nabla^2 L(\nu) = \nabla g(\nu)^T \nabla^2 L \nabla g(\nu) \tag{12}$$

In particular, Dinh et al. (2017) explicitly show, using layer-wise transformations $T_\alpha : (w_1, w_2) \to (\alpha w_1, \alpha^{-1} w_2)$, that deep rectifier feedforward networks possess large numbers of symmetries which can be exploited to control sharpness without changing the network output. The existence of these symmetries in the loss function, under which the geometry of the local loss can be substantially modified (and in particular, the spectral norm and trace of the Hessian) means that the relationship between the local flatness of the loss landscape and generalization is a subtle one.

It's instructive to consider the PAC Bayes generalization bound that motivates SAM, the derivation of which starts from a PAC-Bayesian generalization bound (McAllester, 1999; Dziugaite & Roy, 2017):

**Theorem 1.** *For any distribution $\mathcal{D}$ and prior $p$ over the parameters $w$, with probability $1 - \delta$ over the choice of the training set $\mathcal{S} \sim \mathcal{D}$, and for any posterior $q$ over the parameters:*

$$\mathbb{E}_q[L_{\mathcal{D}}(w)] \leq \mathbb{E}_q[L_{\mathcal{S}}(w)] + \sqrt{\frac{KL(q||p) + \log \frac{n}{\delta}}{2(n-1)}} \tag{13}$$

where the KL divergence:

$$\mathbb{D}_{KL}[q, p] = \mathbb{E}_{p(w)} \left[ \log \left( \frac{q(w)}{p(w)} \right) \right] \tag{14}$$

defines a statistical distance $\mathbb{D}_{KL}[q, p]$ (though not a metric, as it's symmetric only to second order) on the space of probability distributions. Assuming an isotropic prior $p = N(0, \sigma_p^2 I)$ for some $\sigma_p$, an isotropic posterior $q = N(w, \sigma_q^2 I)$, so that $\mathbb{E}_q[L_{\mathcal{D}}(w)] = \mathbb{E}_{\epsilon \sim N(0, \sigma_q^2 I)}[L_{\mathcal{D}}(w + \epsilon)]$, applying the covering approach of Langford & Caruana (2001) to select the best (closest to $q$ in the sense of KL divergence) from a set of pre-defined data-independent prior distributions satisfying the PAC generalization bound, Foret et al. (2020) show that the bound in theorem 1 can be written in the following form:

$$\mathbb{E}_{\epsilon \sim N(0, \sigma_q^2 I)}[L_{\mathcal{D}}(w + \epsilon)] \leq \mathbb{E}_{\epsilon \sim N(0, \sigma_q^2 I)}[L_{\mathcal{S}}(w + \epsilon)] + g \left( \frac{\|w\|_2^2}{\rho^2} \right) \tag{15}$$

(for a monotone function $g$). Then, crucially, one may apply a well-known tail-bound for a chi-square random variable to bound $\|\epsilon\|_2$, thus bounding the expectation over $q$ (with probability $1 - 1/\sqrt{n}$) by the maximum value over a Euclidian norm-ball ball. This provides the following generalization bound:

**Theorem 2.** *For any $\rho > 0$ and any distribution $\mathcal{D}$, with probability $1 - \delta$ over the choice of the training set $\mathcal{S} \sim \mathcal{D}$,*

$$L_{\mathcal{D}}(w) \le \max_{\|\epsilon\|_2 \le \rho} L_{\mathcal{S}}(w + \epsilon) + g\left(\frac{\|w\|_2^2}{\rho^2}\right) \tag{16}$$

*where $\rho = \sigma \sqrt{k}\left(1 + \sqrt{\frac{\ln(n)}{k}}\right)$, $n = |\mathcal{S}|$, and $k$ is the number of parameters.*

This bound justifies and motivates the SAM objective:

$$\max_{\|\Delta w\| \le \rho} L(w + \Delta w) + \lambda \|w\|_2^2 \tag{17}$$

and resulting algorithm. Whilst the bound in Theorem 2 suggests that the ridge-penalty should vary with the radius of the perturbation, in practice (Foret et al., 2020) the penalty term is fixed (or simply set to zero) even when different perturbation radii are searched over. Subsequent refinements of SAM (Kim et al., 2022) ignore the ridge penalty term altogether, and the choice of an optimal perturbation radius is what drives the success of the method. It is not clear, however, why this adversarial parameter-space perturbation should help generalization more than evaluating (and approximating) the expectation in the very bound which motivates the SAM procedure in the first place, which would lead instead to an objective (ignoring, for now, the ridge penalty term) of the following form:

$$\mathbb{E}_{\epsilon \sim N(0, \sigma^2 I)}[L_{\mathcal{S}}(w + \epsilon)] \tag{18}$$

Moreover, the worst-case adversarial perturbation used by SAM is likely to be noisier and is also naturally a significantly looser bound than the expectation-based bound.

## 2 Generalized variational posterior

Our starting point is a similar, but more general, optimization objective, which arises in the variational optimization of a generalized posterior distribution, $q$, over the space of probability measures $\mathcal{P}(\Theta)$ on the parameter space $\Theta$ given by (Bissiri et al., 2016):

$$q^*(w) = \arg\min_{q \in \mathcal{P}(\Theta)} \left\{\mathbb{E}_{q(w)}[L(w)] + \mathbb{D}_{KL}[q, p]\right\} \tag{19}$$

for which, when $Z = \int_{\Theta} \exp\left\{-\sum_{i=1}^{n} l(w, x_i)\right\} \pi(\theta) d\theta < \infty$, the solution is given by the generalized posterior:

$$q^*(w) \propto p(w) \prod_{i=1}^{N} \exp\{-l(w, x_i)\} \tag{20}$$

The terms $\exp\{-l(w, x_i)\}$ are to be interpreted as quasi-likelihoods, and for the particular choice $l(w, x_i) = -\log p(x_i|w)$, we recover the standard Bayesian posterior. As this infinite dimensional optimization is in general intractable, it is usual to assume that the posterior belongs to a parametric family $\mathcal{Q} \subset \mathcal{P}$:

$$q^*(w) = \arg\min_{q \in \mathcal{Q}(\Theta)} \left\{\mathbb{E}_{q(w)}[L(w)] + \mathbb{D}_{KL}[q, p]\right\} \tag{21}$$

which, for the choice $l(w, x_i) = -\log p(x_i|w)$, is the same objective (up to a constant factor) as the evidence lower bound (ELBO) used in variational Bayes.

In practice, it is often found that tempering the KL divergence term by a positive factor $\rho < 1$ produces optimal performance, giving rise to:

$$q^*(w) = \arg\min_{q \in \mathcal{Q}(\Theta)} \left\{\mathbb{E}_{q(w)}[L(w)] + \rho \mathbb{D}_{KL}[q, p]\right\} \tag{22}$$

## 2.1 TRACER: flatness-inducing regularization

Ignoring for simplicity the contribution from the prior term (which would correspond to a ridge-regularization term under the assumption $p(w) \sim N(0, \sigma_p I)$), leads to following objective, which we seek to minimize over $w$:

$$\mathbb{E}_{q(w)}[L(w)] - \rho\mathcal{H}(q) \tag{23}$$

where $\mathcal{H}(q) = -\mathbb{E}_{q(w)}[q(w)]$ is the entropy of $q$. For the choice $q(w) \sim N(w, \sigma^2 I)$, the optimization problem associated with the variational objective becomes (absorbing some constants into $\rho$):

$$\arg\min_{q,\Sigma} \mathbb{E}_q[L(w)] - \rho\mathcal{H}(q) = \arg\min_{w,\sigma^2} \mathbb{E}_q[L(w)] + \rho\log\frac{1}{\sigma^2} \tag{24}$$

so that we can see that $\rho$ determines the variance of Gaussian perturbation over which the loss is averaged. More generally, choosing $q \sim N(w, \Sigma)$ leads to the following variational objective:

$$\arg\min_{q,\Sigma} \mathbb{E}_q[L(w)] - \rho\mathcal{H}(q) = \arg\min_{w,\Sigma} \mathbb{E}_q[L(w)] + \rho\log\frac{1}{\det(\Sigma)} \tag{25}$$

so that large values of $\rho$ will correspond to distributions with larger volume, since for $x \sim N(0, \Sigma)$, $x$ lies within the ellipsoid $x^T \Sigma^{-1} x = \chi^2(\alpha)$ with probability $1 - \alpha$, with the volume of the ellipsoid proportional to $\det(\Sigma)^{\frac{1}{2}}$ (Anderson, 2003). We show in Section 2.3, by expanding the expectation under $q$ to second order, i.e.

$$\mathbb{E}_{q(w)}[L(w)] \approx L(w) + \frac{1}{2}\mathrm{Tr}(\Sigma\nabla_w^2 L(w)) \tag{26}$$

that in the neighborhood of a local minimum (where the Hessian is positive-definite) the curvature of the loss-surface is penalized over a region whose volume is determined by $\rho$. While intuitively appealing, this flatness inducing penalty is not invariant to coordinate transformations, so that scale changes (such as occur, for example, when applying batch-normalization or weight-normalization) which have no effect on the output of the learned probability distribution, can nevertheless still result in arbitrary changes to the penalty. More generally any geometric notion of loss surface flatness must be independent of arbitrary rescaling of the network parameters. Motivated by these considerations, we apply steepest descent in the KL-metric (also known as natural gradient descent (Amari, 1998)) to our variational objective, under the assumption $q(w) \sim N(w, \Sigma)$, in order to find solutions to:

$$\arg\min_{\mu,\Sigma} \mathbb{E}_q[L(w)] + \rho\mathbb{D}_{KL}[q, p] \tag{27}$$

where $\rho$ is a positive real-valued regularization parameter.

We show in the sequel that, assuming an isotropic Gaussian prior, $p(w) \sim N(0, \eta I)$, performing gradient descent w.r.t. the natural gradient then leads to the following iterative update equations:

$$\mu \leftarrow \mu - \alpha\Lambda^{-1}\left(\mathbb{E}_q[\nabla_w L(w)] + \frac{\rho}{\eta}w\right) \tag{28}$$

$$\Lambda \leftarrow (1 - \beta)\Lambda + \beta\left(\frac{\mathbb{E}_q[\nabla_w^2 L(w)]}{\rho} + \eta^{-1}I\right) \tag{29}$$

where $\alpha$ and $\beta$ are the learning rates for the mean and precision updates, respectively, and $\Lambda := \Sigma^{-1}$ is the precision matrix. Approximating the expectations to second order and further simplifying leads to the following update equations (see below for a detailed derivation):

$$\mu \leftarrow \mu + \alpha\overline{H}^{-1}\left(\nabla_w[L(w) + \rho\mathrm{Tr}(H\overline{H}^{-1})]\right) \tag{30}$$

$$\overline{H} \leftarrow (1 - \beta)\overline{H} + \beta H \tag{31}$$

where $H = \nabla_w^2 L(w)$ is the Hessian, $\overline{H}$ is an exponential smoothing of the Hessian, and the update for the mean consists of a preconditioned (by the inverse smoothed Hessian) gradient, together with, crucially, a

---

**Algorithm 1** SGD-TRACER
___
**Require:** $\alpha_t$: Stepsize
**Require:** $\beta$: Exponential smoothing constant for the online Fisher estimate
**Require:** $\rho$ : flatness inducing penalty term
**Require:** $\delta$: small positive constant
  Initialize $\mathbf{w}_0$, $\mathbf{f}_0$, $t = 0$
  **while** not converged do **do**
    Sample batch $\mathcal{B} = \{(\boldsymbol{x}_1, \boldsymbol{y}_1), ...(\boldsymbol{x}_b, \boldsymbol{y}_b)\}$
    $\mathbf{w}_{t+1} = \mathbf{w}_t - \alpha_t \nabla_{\mathbf{w}} \left[ L_{\mathcal{B}}(\mathbf{w}_t) + \rho \left\langle (\nabla_{\mathbf{w}} L_{\mathcal{B}}(\mathbf{w}_t))^2, (\bar{\mathbf{f}}_t + \delta)^{-1} \right\rangle \right]$
    $\bar{\mathbf{f}}_{t+1} = (1 - \beta)\bar{\mathbf{f}}_t + \beta (\nabla_w L_{\mathcal{B}}(\mathbf{w}_t))^2$
  **end while**

---

penalty term proportional to the (affine-invariant) ratio of the Hessian and the smoothed Hessian. Finally, via the diagonal Empirical Fisher approximation of the Hessian (see below for details, and a discussion of alternatives), allowing for a dynamic learning rate $\alpha_t$, and dropping the preconditioner, we arrive at a modified SGD-type update which we call SGD-TRACER.

## 2.2 SGD-TRACER

SGD-TRACER is given by Algorithm (1), in which the usual stochastic gradient update is modified with a term which penalizes the trace of the ratio between the diagonal of the Empirical Fisher Information Matrix (FIM) and an exponentially weighted average the of the Empirical FIM diagonal. By augmenting the loss with a TRACER term and maintaining a smoothed squared-gradient estimate, in principle, any optimization scheme can be modified in the same way. In our experiments we use SGD with momentum for vision tasks and Adam-TRACER for NLP tasks, based on standard practice in each problem domain.

## 2.3 Derivation of the TRACER flatness-inducing regularizer

Following (Khan & Rue, 2021) and (Zhang et al., 2017), we make the assumption $q(w) \sim N(\mu, \Sigma)$ and seek to optimize the variational objective in Equation (22) w.r.t. the variational parameters $\phi = (\mu, \Sigma)$ using natural gradient descent. This allows us to derive an algorithm that respects the intrinsic geometry of the parameter space, and thus derive an algorithm that seeks sharp minima in an approximately coordinate-independent way.

Thus we aim to minimize:

$$\mathcal{L}(\phi) := \mathbb{E}_q[L(w)] + \rho \mathbb{D}_{KL}[q, p] \tag{32}$$

where $\rho$ is a positive real-valued regularization parameter. The negative gradient corresponds to the steepest descent direction in the Euclidian metric:

$$\frac{-\nabla_\phi \mathcal{L}}{\|\nabla_\phi \mathcal{L}\|} = \lim_{\epsilon \to 0} \frac{1}{\epsilon} \underset{\Delta\phi:\|\Delta\phi\|_2 < \epsilon}{\text{argmin}} \mathcal{L}(\phi + \Delta\phi) \tag{33}$$

and thus depends on the chosen coordinates $\phi$. In contrast, the so-called natural gradient update corresponds to steepest descent in the KL-divergence metric:

$$\frac{-F^{-1}\nabla_\phi \mathcal{L}}{\|\nabla_\phi \mathcal{L}\|} = \lim_{\epsilon \to 0} \frac{1}{\epsilon} \underset{\Delta\phi:\mathbb{D}_{KL}[q_\phi, q_{\phi+\Delta\phi}] < \epsilon}{\text{argmin}} \mathcal{L}(\phi + \Delta\phi) \tag{34}$$

where $F$ is the FIM:

$$F := \mathbb{E}_{q_\phi(w)} \left[ \nabla_\phi \log q_\phi(w)^T \nabla_\phi \log q_\phi(w) \right] = \mathbb{E}_{q_\phi(w)} \left[ -\nabla_\phi^2 \log q_\phi(w) \right] \tag{35}$$

which defines a Riemannian metric on the parameter manifold $\Phi$ where $\mathcal{Q}(\Theta) = \{q_\phi(w) : \phi \in \Phi\}$. Expanding to second order in a small neighborhood of $\phi$ we have:

$$\mathbb{D}_{KL}[q_\phi, q_{\phi+\Delta\phi}] = \mathbb{E}_{q_\phi(w)} \left[ -\Delta\phi^T \nabla_\phi \log q_\phi(w) - \frac{1}{2}\Delta\phi^T \nabla_\phi^2 \log q_\phi(w)\Delta\phi \right] + O(||\Delta\phi||^3) \tag{36}$$

and since:

$$\mathbb{E}_{q_\phi(w)}\nabla_\phi \log q_\phi(w) = \mathbb{E}_{q_\phi(w)}\left[\frac{\nabla_\phi q_\phi(w)}{q_\phi(w)}\right] = \nabla_\phi \mathbb{E}_{q_\phi(w)}[1] = 0 \tag{37}$$

the FIM (under certain regularity conditions) can be seen to be the Hessian (or curvature) of the K-L divergence:

$$\mathbb{D}_{KL}[q_\phi, q_{\phi+\Delta\phi}] = -\frac{1}{2}\Delta\phi^T \mathbb{E}_{q_\phi(w)}\left[\nabla_\phi^2 \log q_\phi(w)\right]\Delta\phi + O(||\Delta\phi||^3) = \frac{1}{2}\Delta\phi^T F\Delta\phi + O(||\Delta\phi||^3) \tag{38}$$

The following proposition gives an expression for the natural gradient vector (for proof see Appendix A.5):

**Proposition 1.** *For a probability distribution with pdf $q_\phi(w) \sim N(\mu, \Lambda^{-1})$ with the parameterization $\phi = \begin{bmatrix} \mu \\ \text{vec}(\Lambda) \end{bmatrix}$, the natural gradient $\tilde{\nabla}_\phi$ of $\mathcal{L}(\phi)$ is given by:*

$$\tilde{\nabla}_\phi \mathcal{L}(\phi) = \begin{bmatrix} \tilde{\nabla}_\mu \mathcal{L} \\ \text{vec}(\tilde{\nabla}_\Lambda \mathcal{L}) \end{bmatrix} \tag{39}$$

*where*

$$\tilde{\nabla}_\mu \mathcal{L} = \Sigma \mathbb{E}_q[\nabla_w L(w) + \rho \nabla_w p(w)] \tag{40}$$

$$\tilde{\nabla}_{\Sigma^{-1}} \mathcal{L} = -\mathbb{E}_q[\nabla_w^2 L(w) - \rho \nabla_w^2 p(w)] + \rho\Sigma^{-1} \tag{41}$$

Assuming an isotropic Gaussian prior, $p(w) \sim N(0, \eta I)$, performing gradient descent w.r.t. this natural gradient then leads to the following iterative update equations:

$$\mu \leftarrow \mu - \alpha\Lambda^{-1}\left(\mathbb{E}_q[\nabla_w L(w)] + \frac{\rho}{\eta}w\right) \tag{42}$$

$$\Lambda \leftarrow (1-\beta)\Lambda + \beta\left(\frac{\mathbb{E}_q[\nabla_w^2 L(w)]}{\rho} + \eta^{-1}I\right) \tag{43}$$

where $\alpha$ and $\beta$ are the learning rates for the mean and precision updates, respectively. We work with each of these update equations in turn. Starting with the update equation for the mean $\mu$, the key observation is that the expectation $\mathbb{E}_q[\nabla_w L(w)]$ is taken with respect to the distribution $q(w)$, which is an exponential moving average of the expected Hessian $\mathbb{E}_q[\nabla_w^2 L(w)]$. This updating happens naturally as a consequence of taking natural gradient steps, and leads to an approximately coordinate-free algorithm in the sequel. Applying Bonnet's theorem (Khan & Rue, 2021) and forming the second-order approximation to the loss we obtain:

$$\mathbb{E}_q[\nabla_w L(w)] = \nabla_\mu \mathbb{E}_q[L(w)] \approx \nabla_\mu \mathbb{E}_q[L(\mu) + (w-\mu)^T \nabla_w^2 L(w)|_{w=\mu}(w-\mu)] \tag{44}$$

Writing $w - \mu = \Sigma^{\frac{1}{2}}\nu$ where $\nu \sim \mathcal{N}(0, I)$ we have:

$$\mathbb{E}_q[(w-\mu)^T \nabla_w^2 L(w)|_{w=\mu}(w-\mu)] = \mathbb{E}_{\nu\sim\mathcal{N}(0,I)}[\nu^T \Sigma^{\frac{1}{2}^T}\nabla_w^2 L(w)|_{w=\mu}\Sigma^{\frac{1}{2}}\nu] = \text{Tr}(\Sigma^{\frac{1}{2}^T}\nabla_w^2 L(w)|_{w=\mu}\Sigma^{\frac{1}{2}}) = \text{Tr}(H\Sigma) \tag{45}$$

where we used the fact that $\mathbb{E}_{\nu\sim\mathcal{N}(0,I)}[\nu^T \Omega\nu] = \sum_{i,j}\Omega_{i,j}\mathbb{E}[\nu_i\nu_j] = \text{Tr}(\Omega)$, and where $H$ is the Hessian $\nabla_w^2 L(w)$. We therefore have that:

$$\mathbb{E}_q[\nabla_w L(w)] \approx \nabla_\mu[L(\mu) + \text{Tr}(H\Sigma)] \tag{46}$$

Choosing the prior variance $\eta$ to be infinite and thus ignoring terms involving $\eta$ in both update equations (corresponding to an improper prior, and so consistent with the discussion above), leads to the following update for the mean:

$$\mu \leftarrow \mu + \alpha\Lambda\left(\nabla_\mu[L(\mu) + \text{Tr}(H\Sigma)]\right) \tag{47}$$

Thus, in order to blur the loss with multivariate Gaussian noise in a way that aligns with the intrinsic geometry of the parameter space, we can (to second order) augment the loss with a term involving the Trace

of the Hessian. Considering now the update equation for the precision, we can use Price's theorem (Khan & Rue, 2021) together with a Taylor expansion to get, to second order $\mathbb{E}_q[\nabla_w^2 L(w)] \approx \nabla_w^2 L(w)|_{w=\mu}$ (see Appendix A.2 for details), which leads to

$$\Lambda \leftarrow (1-\beta)\Lambda + \beta \left( \frac{\nabla_w^2 L(w)|_{w=\mu}}{\rho} \right) \tag{48}$$

We next substitute, as is common in the literature using approximate second order approximation (Martens, 2020), the Generalized Gauss-Newton matrix (GGN) for the Hessian, given by:

$$G(w) = \frac{1}{|S|} \sum_{(x,y)\in S} [J_f^T H_L J_f] \tag{49}$$

where $J_f$ is the Jacobian of the output function $f$ and $H_L$ is the Hessian of the loss w.r.t. the output distribution. The GGN is a positive definite approximation to the Hessian which converges to the Hessian as the fitted residuals go to zero, (Kunstner et al., 2019). The most practically relevant losses, cross-entropy (classification), and squared error (regression) correspond to exponential family output distributions with natural parameters given by the output function $f(x,w)$, together with for the log-loss $l(y, f(x,w)) = -\log p(y|x, w)$. For these choices, the GGN is equivalent to the Fisher Information Matrix. While the evaluation of the GGN matrix, in particular the matrix multiplications involving the Jacobians $J_f$, can be relatively costly, the FIM can be expressed as an expectation of outer products of gradients w.r.t. the output distribution $p(y|x, w)$:

$$\frac{1}{n} \sum_{i=1}^n \mathbb{E}_{p(y|x_i, w)} \left[ \nabla_w \log p(y|x_i, w)^T \nabla_w \log p(y|x_i, w) \right] \approx \frac{1}{n} \sum_{i=1}^n \nabla_w \log p(\tilde{y}_i|x_i, w)^T \nabla_w \log p(\tilde{y}_i|x_i, w) := \tilde{F} \tag{50}$$

which, following Martens (2020), can be estimated using a single Monte Carlo sample from the output distribution: $\tilde{y} \sim p(y|x_i, w)$. Using this (biased) Fisher approximation in our setting thus requires gradients to be calculated through an expectation $\nabla_w \mathbb{E}_{p(y|x, w)}[L(w; y)]$, approximated using a Monte Carlo sample from the model's output distribution. Since the expectation is taken w.r.t. a distribution which depends on $w$, it is necessary to reparameterize so that the discrete Monte Carlo sample is expressed as the deterministic transformation of a $g_w(z)$ (depending on $w$) of a sample $z \sim h_\theta(z)$ from a distribution not depending on $w$, so that $\mathbb{E}_{p(y|x, w)}[L(w; y)] = \mathbb{E}_{z \sim h_\theta(z)}[L(w; g_w(z))]$. In the discrete case (corresponding to classification), since the argmax function is non-differentiable, the standard approach is the Gumbel-Softmax reparameterization (Jang et al., 2016), which uses the softmax function as a continuous relaxation of the argmax function together with i.i.d. samples distributed as Gumbel(0,1).

It's important to note that this approach is different from simply evaluating $\log p(y|x, w)$ on the training labels, a widely-used approximation known as the empirical Fisher $F_{\text{emp}}$:

$$F_{\text{emp}} := \sum_{i=1}^n \nabla_w \log p(y_i|x_i, w)^T \nabla_w \log p(y_i|x_i, w) \tag{51}$$

This, despite lacking the same convergence guarantees, performs competitively in many settings (Kunstner et al., 2019). We find in our experiments that the empirical Fisher performs competitively with the MC approximation to the GGN (Khan et al., 2018; Kingma & Ba, 2014) and has the advantage of being straightforward and cheap to compute from already computed gradients (in the case of Adam-TRACER, the smooth squared gradients are already computed and maintained for use as a preconditioner). Given the conceptual and computational simplicity of this approach we substitute the empirical Fisher for the Hessian. Recent advances in approximate second-order methods in optimization, notably Yao et al. (2020), suggest avenues for improvement, and we leave investigations of alternatives, such as the smoothed (Hessian-free) Hessian diagonal sketch used in AdaHessian, for future work.

Substituting the empirical Fisher approximation for the Hessian in the update equation for the precision 48, and rewriting the update in terms of $\overline{F} := \rho\Lambda$, absorbing a factor $\rho$ in to $\alpha$, and writing the iteration in

terms of the parameter $w$, we obtain:

$$w \leftarrow w + \alpha \overline{F}^{-1} \left( \nabla_w [L(w) + \rho \text{Tr}(F\overline{F}^{-1})] \right) \tag{52}$$

$$\overline{F} \leftarrow (1 - \beta)\overline{F} + \beta F \tag{53}$$

Crucially, the penalty term $\rho \text{Tr}(F\overline{F}^{-1})$ can be seen to be invariant to affine coordinate transformations, since it is the trace of the ratio of two (0,2) tensors which transform in the same way. Indeed, under an affine coordinate transformation with Jacobian $J$, we have $F \rightarrow J^T F' J$ and $\overline{F} \rightarrow J^T \overline{F}' J$ so that:

$$\text{Tr}\left( F\overline{F}^{-1} \right) = \text{Tr}\left( J^T F' J J^{-1} \overline{F}'^{-1} J^{T-1} \right) = \text{Tr}\left( J^{T-1} J^T F' \overline{F}'^{-1} \right) = \text{Tr}\left( F'\overline{F}'^{-1} \right) \tag{54}$$

By using the ratio of the (squared) gradients and the exponentially smoothed gradients, the trace ratio in effect penalizes the change in (squared) gradient, in a coordinate-free way. More generally, given a smooth coordinate change defined by a diffemorphism $\Phi : \mathbb{R}^p \rightarrow \mathbb{R}^p$ and Jacobian $J(w)$, then given sufficiently rapid exponential decay in the update equation for the Fisher, subject to $\Phi$ having sufficient regularity, the penalty term is readily seen to be approximately coordinate free.

We now make two simplifications. First, we use a mean-field approximation, representing the FIM by its diagonal, as is done in Adam (Kingma & Ba, 2014) and Adagrad (Duchi et al., 2011), thus:

$$F \approx \frac{1}{n} \sum_{i=1}^{n} \nabla_w \log p(y_i|x_i, w)^2 \tag{55}$$

Secondly, it is standard practice to (Martens, 2020) to add Tikhonov regularization or damping via a small positive real constant $\delta$ when using 2nd-order optimization methods, giving in this case the preconditioner: $(\overline{F} + \delta I)^{-1}$. In fact this would arise naturally in our setup by choosing $\eta$ to be non-zero, in which case we would simply have $\delta := \frac{\rho}{\eta}$. From an optimization perspective, it is justified by recognizing that the local quadratic model from which the second-order update is ultimately derived is a second-order approximation to the KL divergence and is thus only valid locally. For directions corresponding to small eigenvalues, parameter updates can lie outside the region where the approximation is reasonable (Martens, 2020). This is true, a fortiori, when diagonal approximations are used, as is the case here. As our emphasis here is on geometric regularization, we drop the preconditioner entirely by choosing $\delta$ to be sufficiently large that the preconditioner is equal to the identity (up to a constant, which is absorbed into the learning rate).

Finally, as most current deep learning frameworks don't straightforwardly support access to per-example gradients, which can in principle be achieved with negilible additional cost (see, for example, BackPACK Dangel et al. (2020) second-order Pytorch extensions), for simplicity and efficiency, we use the gradient magnitude (GM) approximation (Bottou et al., 2016), as used in standard optimizers Adam and RMSprop, replacing the sum of squared gradients with the square of summed gradients:

$$\frac{1}{n} \sum_{i=1}^{n} [\nabla_w \log p(y_i|x_i, w)]^2 \approx \left[ \frac{1}{n} \sum_{i=1}^{n} \nabla_w \log p(y_i|x_i, w) \right]^2 \tag{56}$$

Writing the resulting FIM diagonal as $(\nabla_w L(w))^2$, we finally end up with the following simple update equations:

$$w_{t+1} = w_t - \alpha_t \nabla_w \left[ L(w_t) + \rho \left\langle (\nabla_w L(w_t))^2, (\overline{f}_t + \delta)^{-1} \right\rangle \right]$$
$$\overline{f}_{t+1} = (1 - \beta)\overline{f}_t + \beta (\nabla_w L(w_t))^2 \tag{57}$$

which are summarized in Algorithm 1. We show in appendix A.3 that the algorithm converges to a neighborhood of a local minimum of $L(w)$ of size $\mathcal{O}(\rho^2)$. We note in passing that, in this simplest form (after applying the gradient magnitude approximation), the update equations amount to regularizing with a (scale-adjusted) gradient norm. In principle (particularly for the large batch case) we would expect to see significant improvements by moving to per-gradient calculations (which are theoretically no more expensive to compute, but require additional work under most current ML frameworks).

### 2.4 Results

We first examine a challenging variant on a standard benchmark in computer vision, CIFAR-100. We compare SGD, SAM and SGD-Tracer using none of the standard regularizations (no data augmentation, no weight-decay) and a standard training protocol (200 epochs, initial learning rate set to 0.1, cosine learning-rate decay). Further, we randomly flip 50% of the labels so that 50% of examples are incorrectly labeled. The results in Table 3 show that GTRACER significantly improves on SAM in this challenging setting. In Figure 1 we highlight results for the same problem over different values of the regularization parameter $\rho$. In Figure 2 we compare the training curves on this problem.

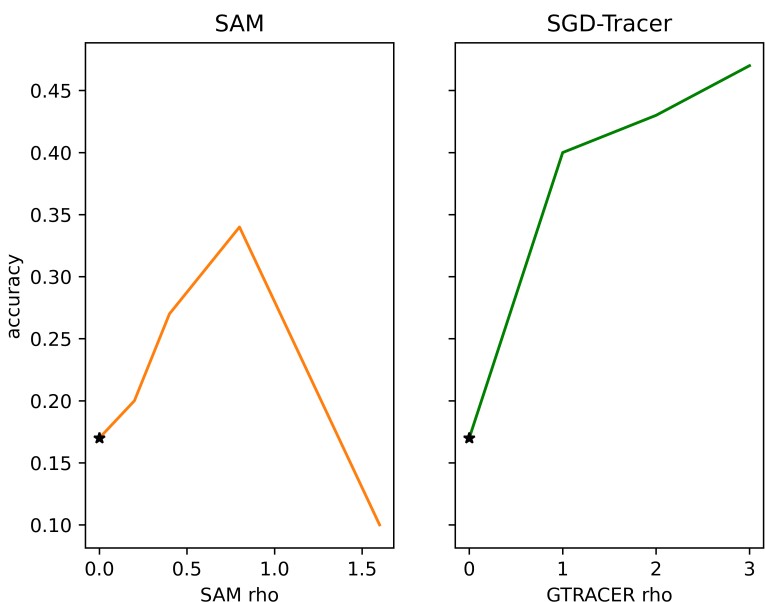

Figure 1: CIFAR 100: ResNet20, no weight-decay, 50% noise, accuracy vs regularization strength. GTRACER dominates the baseline and SAM across a wide range of regularization strengths.

Table 1: CIFAR 100: ResNet20, no weight-decay, 50% noise, accuracy (standard error)

|  | no aug |
|---|---|
| SGD | 17.5% (2.41) |
| SAM | 34.63% (1.85) |
| SGD-TRACER | **47.55%** (1.51) |

We next run SGD-Tracer on CIFAR-100 with and without label noise, with and without augmentation, with random label flipping and with a standard ridge penalty of $5 \times 10^{-4}$. The results in Table 2 show that SGD-TRACER performs consistently well, with a particularly strong advantage in the the presence of noise and/or without additional regularization in the form of data augmentation. For NLP tasks we use

Table 2: CIFAR-100: ResNet20, accuracy (standard error)

|  | no aug | with aug | 50% noise & no aug |
|---|---|---|---|
| SGD | 51.43 % (0.41) | 70.02% (0.36) | 21.96% (0.36) |
| SAM | 58.98 % (0.52) | 70.33% (0.22) | 49.89% (0.32) |
| SGD-TRACER | **63.47%** (0.32) | **70.71%** (0.36) | **51.62%** (0.18) |

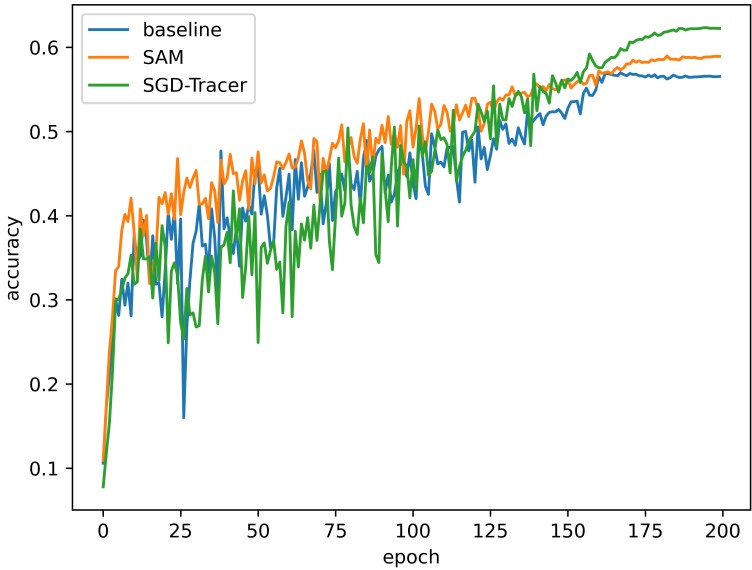

Figure 2: CIFAR 100: ResNet20, 50% noise, test-accuracy training curves. On a standard 200 epoch training protocol with cosine learning-rate decay, SGD-Tracer converges to a solution that generalizes better than SGD and SAM

the Huggingface Bert-base-uncased checkpoint together with Adam-TRACER. We fine-tune using Adam-Tracer, using a standard protocol of 5 epochs with initial learning rate $2 \times 10^{-5}$. Each run is repeated 20 times. Performance is uniformly strong across the 3 benchmark tasks (taken from the challenging SuperGlue benchmark), and Adam-TRACER has the additional property of producing more stable results across runs (as reflected in the standard errors). See the Appendix for details of (standard) experiment hyperparameters.

Table 3: SupeGlue tasks BERT base-uncased results, accuracy (standard error)

|  | **BOOLQ** | **WIC** | **RTE** |
|---|---|---|---|
| Adam | 73.84% (0.14) | 69.36% (0.08) | 69.18% (0.33) |
| SAM | 73.95% (0.13) | 69.06% (0.07) | 69.54% (0.28) |
| Adam-TRACER | **75.09%** (0.04) | **70.01%** (0.06) | **70.13%** (0.18) |

## 3 Conclusion

Motivated by the notable empirical success of SAM, a prior that flat (in expectation, and in an intrinsic, geometric sense) minima should generalize better than sharp minima, and noting the connections between the generalized Bayes objective and SAM, we have derived a new algorithm that is simple to implement and understand, cheap to evaluate, provably convergent, naturally scale-independent (and approximately coordinate-free) and which is competitive with SAM on key benchmark problems. Performance is particularly strong for challenging low signal-to-noise ratio and large batch problems. Crucially the algorithm is straightforwardly derived from an approximate natural gradient optimization of an ELBO-type objective and doesn't rely on "m-sharpness" (Foret et al., 2020) or other poorly understood (and expensive to compute) heuristics.

# A  Appendix

## A.1  Multivariate Gaussian Fisher Information Matrix, $[\mu, \mathrm{vec}(\Lambda)]^T$ parameterization

For a probability distribution with density $q$ with parameters $\phi$, the Fisher Information Matrix (FIM) can be written as the expected negative log-likelihood Hessian:

$$F = \mathbb{E}_q\left[-\nabla_\phi^2 \log q\right] \tag{58}$$

In particular, for a multivariate Gaussian with pdf: $q(x) \sim N(\mu, \Lambda^{-1})$, parameterized by $\phi = \begin{bmatrix} \mu \\ \mathrm{vec}(\Lambda) \end{bmatrix}$ the negative log-likelihood is, up to constant terms:

$$-\log q(x) = \frac{1}{2}(x-\mu)^T\Lambda(x-\mu) + \frac{1}{2}\log|\Lambda^{-1}| \tag{59}$$

Taking gradients w.r.t. $\mu$, we have: $-\nabla_\mu \log q(x) = \Lambda(x-\mu)$ and therefore $\mathbb{E}\left[\nabla_\mu^2 q(y)\right] = \Lambda$. Taking gradients w.r.t. the covariance, and since $\nabla_\Lambda (x-\mu)^T\Lambda(x-\mu) = (x-\mu)(x-\mu)^T$ and $\nabla_\Lambda \log|\Lambda^{-1}| = \nabla_\Lambda \log|\Lambda|^{-1} = -\nabla_\Lambda \log|\Lambda| = -(\Lambda^T)^{-1} = -(\Lambda)^{-1}$ we have:

$$\nabla_\Lambda q(x) = \frac{1}{2}(x-\mu)(x-\mu)^T - \frac{1}{2}\Lambda^{-1} \tag{60}$$

Finally, writing $\nabla_\Lambda \Lambda^{-1}$ as $-\Lambda \otimes \Lambda$ and $\Lambda^{-1} := \Sigma$ we have:

$$\nabla_\Lambda^2 q(x) = \frac{1}{2}\Sigma \otimes \Sigma \tag{61}$$

so that the FIM is given by:

$$F = \mathbb{E}_q\left[-\nabla_\phi^2 \log q\right] = \begin{bmatrix} \Sigma^{-1} & 0 \\ 0 & \frac{1}{2}\Sigma \otimes \Sigma \end{bmatrix} \tag{62}$$

## A.2  Approximate expected Hessian

**Lemma 1.** *To second order, we can approximate the expected Hessian w.r.t. a multivariate Gaussian with pdf: $q(x) \sim N(\mu, \Lambda^{-1})$ by its value at the mean:*

$$\mathbb{E}_q[\nabla_w^2 L(w)] \approx \nabla_w^2 L(w)|_{w=\mu} \tag{63}$$

*Proof.* Following Khan & Rue (2021), by Price's theorem, we have:

$$\mathbb{E}_q[\nabla_w^2 L(w)] = 2\nabla_{\Lambda^{-1}}^2 \mathbb{E}_q[L(w)] \tag{64}$$

expanding the r.h.s. to second order using a Taylor series, this is equivalent to:

$$2\nabla_{\Lambda^{-1}}^2 \mathbb{E}_q[(w-\mu)^T\nabla_w^2 L(w)|_{w=\mu}(w-\mu)] \tag{65}$$

Finally, noting that $\mathbb{E}_q[(w-\mu)^T\nabla_w^2 L(w)|_{w=\mu}(w-\mu)] = \mathrm{Tr}\left[\frac{1}{2}\Lambda^{-1}\nabla_w^2 L(w)|_{w=\mu}\right]$, we have, to second order:

$$\mathbb{E}_q[\nabla_w^2 L(w)] \approx 2\nabla_{\Lambda^{-1}}^2 \mathrm{Tr}\left[\frac{1}{2}\Lambda^{-1}\nabla_w^2 L(w)|_{w=\mu}\right] = \nabla_w^2 L(w)|_{w=\mu} \tag{66}$$

$\square$

### A.3 Convergence analysis

With $T(w_t) := \left\langle (\nabla_w L(w_t))^2, (\overline{f} + \delta)_t^{-1} \right\rangle$, as $\rho \to 0$, the iterates $w_{t+1} = w_t - \alpha_t \nabla_w [L(w_t) + \rho \nabla_w T(w_t)]$ will converge to those of SGD. For $\rho > 0$, the algorithm is biased away from a pure descent direction, and convergence then depends on the magnitude of $\rho$. The key assumption in the following convergence proof is that $\|\rho \nabla_w T(w_t)\|_2^2 \leq \kappa \|\nabla_w L(w_t)\|_2^2 + \zeta$ , which controls the bias. This follows from the standard assumption of twice-differentiability of $L(w)$ and the Lipschitz continuity of $\nabla_w L(w_t)$, which imply that the Hessian has a bounded spectral norm:

$$\|\rho \nabla_w T(w_t)\|_2^2 \leq 4\rho^2 \|\nabla_w^2 L(w_t)\|_2^2 \|(\overline{f} + \delta)_t^{-1}\|_2^2$$
$$\leq 4\left(\frac{\rho}{\delta}\right)^2 C^2 p \tag{67}$$

so that $\zeta$ depends on the Lipshitz constant $C$ and the ratio $\frac{\rho}{\delta}$.

**Theorem 3.** *Let $T(w_t) := \left\langle (\nabla_w L(w_t))^2, \overline{f}_t^{-1} \right\rangle$, and assume the objective (loss) $L : \mathbb{R}^p \to \mathbb{R}$ is Lipschitz continuous, twice differentiable, and has Lipschitz-continuous gradient. Let us assume, following Bottou et al. (2016) and Ajalloeian & Stich (2021) that we have a stochastic direction $g(w_t, \xi_t)$ which has the following properties, $\forall t$:*

$$\mathbb{E}[g(w_t, \xi_t)] = \nabla_w L + \rho \nabla_w T(w_t) \tag{68}$$

*and further assuming that there exist $M$, $M_G$ such that, $\forall t$,*

$$\mathbb{E}\left[\|g(w_t, \xi_t)\|^2\right] \leq M + M_G \|\nabla_w L + \rho \nabla_w T(w_t)\|^2 \tag{69}$$

*and the following bound on the bias:*

$$\|\rho \nabla_w T(w_t)\|^2 \leq \kappa \|\nabla_w L(w_t)\|_2^2 + \zeta \tag{70}$$

*then the iteration:*

$$w_{t+1} = w_t - \alpha_t \nabla_w [L(w_t) + \rho \nabla_w T(w_t)]$$
$$\overline{f}_{t+1} = (1 - \beta)\overline{f}_t + \beta (\nabla_w L(w_t))^2 \tag{71}$$

*converges to a neighborhood of a stationary point with $\|\nabla L(w)\|_2^2 = \mathcal{O}(\zeta)$.*

*Proof.* By the Lipschitz continuity of the objective function we have the quadratic bound:

$$L(y) \leq L(x) + \langle \nabla_w L(x), y - x \rangle + \frac{C}{2}\|y - x\|^2 \tag{72}$$

By the quadratic upper bound, the iterates generated by the algorithm satisfy:

$$L(w_{t+1}) - L(w_t) \leq -\alpha_t \langle \nabla_w L(w_t), g(w_k, \xi_k) \rangle + \frac{1}{2}\alpha_t^2 C \|g(w_k, \xi_k)\|_2^2 \tag{73}$$

Taking expectations and applying the variance bound we have:

$$\mathbb{E}L(w_{t+1}) - L(w_t) \leq -\alpha_t \|\nabla_w L(w_t)\|^2 - \alpha_t \rho \nabla_w L(w_t)^T \nabla_w T(w_t) + \frac{1}{2}\alpha_t^2 C \mathbb{E}\left[\|g(w_k, \xi_k)\|_2^2\right]$$

$$= -\alpha_t \|\nabla_w L(w_t)\|^2 - \alpha_t \rho \nabla_w L(w_t)^T \nabla_w T(w_t) + \frac{1}{2}\alpha_t^2 C \left[M + M_G \|\nabla_w L(x) + \rho \nabla_w T(w_t)\|_2^2\right]$$

$$= -\alpha_t \|\nabla_w L(w_t)\|^2 - \alpha_t(1 - \alpha C M_G)\rho \nabla_w L(w_t)^T \nabla_w T(w_t) + \frac{1}{2}\alpha_t^2 C M + \frac{1}{2}\alpha_t^2 C M_G \left(\|\nabla_w L(x)\|_2^2 + \rho \|\nabla_w T(w_t)\|_2^2\right) \tag{74}$$

So that, choosing $\alpha_t < \frac{1}{CM_G}$ and applying the bound on $\|\nabla_w T(w_t)\|$ we have:

$$\mathbb{E}L(w_{t+1}) - L(w_t) \leq -\frac{1}{2}\alpha_t \|\nabla_w L(w_t)\|^2 + \frac{1}{2}\alpha_t^2 C M + \frac{1}{2}\alpha_t \|\rho \nabla_w T(w_t)\|_2^2$$

$$\leq -\frac{1}{2}\alpha_t(1 - \kappa)\|\nabla_w L(w_t)\|^2 + \frac{1}{2}\alpha_t^2 C M + \frac{\alpha_t}{2}\zeta \tag{75}$$

Taking the total expectation, for a fixed $\alpha$, we then have:

$$L_{inf} - L(w_1) \leq \mathbb{E}\left[L(w_{K+1})\right] - L(w_1) \leq -\frac{1}{2}\alpha(1-\kappa)\sum_{t=1}^{K}\|\nabla_w L(w_t)\|^2 + \frac{1}{2}K\alpha^2 CM + \frac{K\alpha}{2}\zeta \quad (76)$$

Finally, we have that:

$$\frac{1}{K}\sum_{t=1}^{K}\|\nabla_w L(w_t)\|^2 = \frac{\alpha CM}{1-\kappa} + 2\frac{F(w_1) - F_{inf}}{K\alpha(1-\kappa)} \xrightarrow{K\to\infty} \frac{\alpha CM}{1-\kappa} + \frac{\zeta}{1-\kappa} \quad (77)$$

$\square$

### A.4 Objective function gradient

**Lemma 2.** *The gradient of the objective 32 towards* $\phi' = \begin{bmatrix} \mu \\ \text{vec}(\Sigma) \end{bmatrix}$ *is given by:*

$$\nabla_\mu \mathcal{L} = \mathbb{E}_q[\nabla_w L(w) - \rho\nabla_w \log p(w)] \quad (78)$$

$$\nabla_\Sigma \mathcal{L} = \frac{1}{2}\mathbb{E}_q[\nabla_w^2 L(w) - \rho\nabla_w^2 \log p(w)] - \frac{\rho}{2}\Sigma^{-1} \quad (79)$$

*Proof.* Taking the negative gradient of the objective wrt to $\mu$, and applying Bonnet's theorem (Khan & Rue, 2021), and the fact that the expectation of the score is 0, we have:

$$\nabla_\mu\left(\mathbb{E}_q[L(w)] + \rho\mathbb{D}_{KL}[q(w), p(w)]\right) = \mathbb{E}_q[\nabla_w L(w)] - \rho\mathbb{E}_q\left[\nabla_w \log p(w)\right] \quad (80)$$

Taking the gradient w.r.t. $\Sigma$, and applying Price's theorem, we have:

$$\nabla_\Sigma\left(\mathbb{E}_q[L(w)] + \rho\mathbb{D}_{KL}[q(w), p(w)]\right) = \frac{1}{2}\mathbb{E}_q\left[\nabla_w^2 L(w) + \rho\nabla_w^2 \log q(w) - \rho\nabla_w^2 \log p(w)\right] \quad (81)$$

and since:

$$\mathbb{E}_q\left[\nabla_w^2 \log q(w)\right] = -\frac{1}{2}\mathbb{E}_q\left[\nabla_w^2\left(\log|\Sigma| + (w-\mu)^T\Sigma^{-1}(w-\mu)\right)\right] = -\Sigma^{-1} \quad (82)$$

We obtain

$$\nabla_\mu \mathcal{L} = \mathbb{E}_q[\nabla_w L(w) - \rho\nabla_w \log p(w)] \quad (83)$$

$$\nabla_\Sigma \mathcal{L} = \frac{1}{2}\mathbb{E}_q[\nabla_w^2 L(w) - \rho\nabla_w^2 \log p(w)] - \frac{\rho}{2}\Sigma^{-1} \quad (84)$$

$\square$

### A.5 Objective function natural gradient

**Proposition 2.**

$$\tilde{\nabla}_\mu \mathcal{L} = \Sigma\mathbb{E}_q[\nabla_w L(w) + \rho\nabla_w p(w)] \quad (85)$$

$$\tilde{\nabla}_{\Sigma^{-1}} \mathcal{L} = -\mathbb{E}_q[\nabla_w^2 L(w) - \rho\nabla_w^2 p(w)] + \rho\Sigma^{-1} \quad (86)$$

*Proof.* By Lemma 2, the gradients $\nabla_{\phi'}$ of the objective $\mathcal{L}(\phi)$ w.r.t. $\phi' = \begin{bmatrix} \mu \\ \text{vec}(\Sigma) \end{bmatrix}$ are given by:

$$\nabla_\mu \mathcal{L} = \mathbb{E}_q[\nabla_w L(w) - \rho\nabla_w \log p(w)] \quad (87)$$

and

$$\nabla_\Sigma \mathcal{L} = \frac{1}{2}\mathbb{E}_q[\nabla_w^2 L(w) - \rho\nabla_w^2 \log p(w)] - \frac{\rho}{2}\Sigma^{-1} \quad (88)$$

The gradient $\tilde{\nabla}_\mu \mathcal{L}$ then follows immediately from the definition of the natural gradient operator. Using the chain rule for matrix derivatives we can also show that:

$$\nabla_\Lambda \mathcal{L} = -\Lambda^{-1} \nabla_\Sigma \Lambda^{-1} \tag{89}$$

So that

$$\nabla_\Lambda \mathcal{L} = -\frac{1}{2} \Lambda^{-1} \mathbb{E}_q [\nabla_w^2 L(w) - \rho \nabla_w^2 \log p(w)] \Lambda^{-1} + \frac{\rho}{2} \Lambda^{-1} \tag{90}$$

Thus, the gradients $\nabla_\phi \mathcal{L}(\phi) = \begin{bmatrix} \nabla_\mu \mathcal{L} \\ \text{vec}(\nabla_\Lambda \mathcal{L}) \end{bmatrix}$ are given by:

$$\begin{bmatrix} \mathbb{E}_q[\nabla_w L(w) - \rho \nabla_w p(w)] \\ -\frac{1}{2} \Lambda^{-1} \mathbb{E}_q[\nabla_w^2 L(w) - \rho \nabla_w^2 \log p(w)] \Lambda^{-1} + \frac{\rho}{2} \Lambda^{-1} \end{bmatrix} \tag{91}$$

The Fisher Information Matrix is given by equation (62): :

$$F = \mathbb{E}_{q_\phi} \left[ -\nabla_\phi^2 \log q_\phi \right] = \begin{bmatrix} \Sigma^{-1} & 0 \\ 0 & \frac{1}{2} \Sigma \otimes \Sigma \end{bmatrix} \tag{92}$$

and therefore

$$F^{-1} \nabla_\phi \mathcal{L}(\phi) = \begin{bmatrix} \Lambda^{-1} & 0 \\ 0 & 2\Lambda \otimes \Lambda \end{bmatrix} \begin{bmatrix} \nabla_\mu \mathcal{L} \\ \text{vec}(\nabla_\Lambda \mathcal{L}) \end{bmatrix} = \begin{bmatrix} \Lambda^{-1} \nabla_\mu \mathcal{L} \\ \text{vec}(2\Lambda \nabla_\Lambda \mathcal{L} \Lambda) \end{bmatrix} \tag{93}$$

where we used the identities $(B^T \otimes A)\text{vec}(X) = \text{vec}(AXB)$ and $(A \otimes B)^{-1} = A^{-1} \otimes B^{-1}$. Since $\text{vec}(2\Lambda \nabla_\Lambda \mathcal{L} \Lambda) = \text{vec}(-\mathbb{E}_q[\nabla_w^2 L(w) - \rho \nabla_w^2 p(w)] + \rho \Lambda)$, we have the required updates. $\qquad \square$

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
