# OpenReview forum: "G-TRACER: Expected Sharpness Optimization"
_TMLR — Rejected by TMLR_

### Review · Reviewer_XL2x · 2024-01-18

**Summary Of Contributions:**

Motivated by the popular wisdom that flat minima usually have better generalization performance, this paper proposes a regularization method that aims to decrease the sharpness (increase the flatness) of the loss at the global/local minima it converges to. The paper started with penalizing the expected loss function by a KL divergence between the prior and posterior distributions of weights. Then, with certain assumptions and simplifications, the theoretical analysis arrived at an algorithmic-level regularization method that requires a computation of the Hessian of loss function. The proposed algorithm is numerically tested on a few simple tasks, which shows that it outperforms the SAM and SGD/Adam in terms of the test error.

**Audience:**

Yes

**Broader Impact Concerns:**

no concerns

**Claims And Evidence:**

No

**Requested Changes:**

see the comments in the weaknesses section

**Strengths And Weaknesses:**

Strength

1: According to the provided experimental results, the proposed regularization method is quite effective, and works better than SAM – the commonly implemented algorithm that reduces loss sharpness.

2: The theoretical derivation provides a lot of intuition on the design of the proposed method/algorithm. On one hand, it makes the design look reasonable and convincing (at least on certain simple settings). On the other hand, the insights behind the derivation is also inspiring.


Weaknesses

1: One major limitation of the proposed method is the requirement of computation of the Hessian of loss function at each step, as well as an inverse of Hessian which is even more computationally costly. This makes the method not as scalable as other methods, such as the SAM. Especially in modern machine learning problems, the models often have a very large number of parameters (as also described in page 2 of the paper), which makes the Hessian matrix super big.

2: The theoretical derivation is largely based on a strong assumption that the posterior $q(w)$ is Gaussian distributed. However, in practice, the posterior is unlikely to have such a well behaved distribution (especially for $w$ that is far from $w^*$). Therefore, although the derivation provides some intuition, it is far from clear that such a derivation, as well as the resulting algorithm/method that the paper proposes, still apply to practical cases. Particularly, does the same intuition generalize to practical cases?

3: The paper only provides experimental evaluation of the proposed algorithm on two relatively simple tasks. Moreover, the paper does not provide a detailed description of the experimental setup, which makes the result harder to judge. Given the strong assumption in derivation (see Weakness 2) and the simple experiments, I feel it is not convincing enough to say the proposed method is superior to SAM in terms of generalization performance (although it looks very promising). In the experiment, I am eager to see more evaluation across different tasks.

4: The paper started with adding a KL divergence between the prior and posterior distributions as the regularization term. However, I don’t see the connection between this KL divergence and the sharpness/flatness of the loss. Therefore, I don’t fully understand why penalizing a KL divergence leads to a flatter minimum.

The above are my major concerns. I also see the paper has a few minor weaknesses, as I describe below.

5: I could not see why adding label noise in the CIFAR-100 experiments. Label noise seems not related to the sharpness/flatness of the loss function, or at least not described in the paper.

6: There are a few incorrect or inaccurate minor claims.

a: In Eq.(3) and later, the paper assumed a finite or countably many local maxima for “modern DNNs”. However, many such DNNs are actually over-parameterized (also admitted by the authors). In fact, for over-parameterized models, the local maxima (or minima) form one or more manifolds, which are continuous (see for example [1]).

b: In the over-parameterized cases, $H$ is low rank, and $det(H)$ should be zero. Then the Gaussian curvature $K$ becomes zero, even though the loss is not flat.

c: I think the claim “in a typical high-dimensional setting, the effect of small differences in curvature (or flatness) is exponentially magnified” (page 2) is not correct, or at least not accurate. This is because the “curvature” is defined as $K=det(H)$, not an individual value in the Hessian $H$. Hence, the example below $H’=(1-\epsilon)H$ cannot be considered as a small difference in curvature. The author misinterpreted a small $\epsilon H$ as a small difference in curvature $det(H)$.

7: [typo] In Eq.(13), the KL divergence is denoted as $KL(q||p)$, however later on it is denoted as $D_{KL}[q,p]$.




[1]: Liu. et. al. Loss landscapes and optimization in over-parameterized non-linear systems and neural networks. Applied and Computational Harmonic Analysis, 2022

---

> ### Author Response · Authors · 2024-04-10
> **Rebuttal pt. 1**
>
> We thank the reviewer for an extremely thorough and thought-provoking review.  We address all the concerns below and in the manuscript which has been thoroughly reworked to address these and other comments, and which we believe is much improved as a result.
>
> 1. The proposed method uses a diagonal empirical Fisher (EF) approximation to the Hessian, as used by Adam, RMSProp, and Adagrad, among others.  Moreover, as the sum of squared gradients, it is available for free.  The additional cost vs SGD is an extra forward and backward pass, which is also the case for SAM, which entails an additional forward and backward pass to compute the worst-case perturbation (inner maximization).  We have applied the method to realistic architectures (e.g. bert-large-cased with 340M parameters) and have not observed scalability issues.  As originally written, our derivation of the method used the full Hessian and introduced the EF at a late stage.  We fully agree that it would be helpful to the audience to highlight the final simple version of the algorithm earlier on.  We also believe there is value in highlighting the method in full generality, since there are many ways of approximating inverse Hessians (eg KFAC [1]) and this is an active area of research.  Moreover, there is exciting work in many frameworks (see, e.g., [2]) allowing processing of per-example gradients (which is in principle no more expensive than computing minibatch gradients) which should open up the way to even stronger performance in the future.
>
> 2. The derivation of SAM relies on assuming an isotropic Gaussian prior and posterior.  This strong assumption, which we relax, is the source of the non-invariance of the SAM algorithm (since a Euclidean norm-ball is used to bound the expectation over an isotropic posterior), and is the reason why SAM isn't a flatness penalty in a meaningful geometric sense.  We generalize this significantly by approximating to second order (rather than to first order) the expectation over a multivariate non-isotropic Gaussian which is, crucially, learnt from the data, and leads to an approximately coordinate-free algorithm.  The assumption of multivariate Gaussian posteriors is one that is central to much of contemporary Bayesian inference (eg variational Bayes).  From this perspective, therefore, our scheme is at least as generally applicable as SAM.  Moreover, from an approximation-theoretic point of view, for a sufficiently deep (at least 2 layers) and wide DNN, even a diagonal (mean field) covariance approximation is capable of approximating any posterior distribution over function values (see, e.g. [3]).  In essence, for such sufficiently deep and wide neural networks as are used in real applications, the universal approximation theorem together with the probability integral transform guarantee that mean-field variational inference is not overly restrictive in practice.  Separately, just as L2-regularization can be derived by (restrictively) assuming an isotropic Gaussian prior on the parameters, but is generally applicable and extremely effective, our method can be seen as Bayes-inspired in the same way.
>
> 3. While our paper is primarily theory-driven, we have significantly reworked the experiments and results section, added additional experiments and results on vision transformers (ViTs) and other architectures, and established consistency with results in the literature.  Although we have grouped them together, the 3 tasks in NLP are distinct, intricate, and challenging (they are incorporated into the challenging SuperGLUE benchmark).  Similarly, the baseline and noisy CIFAR-100 problems are distinct and influence the relative advantages of each method.  We have updated this section to reflect additional experiments and the diversity of tasks already performed, together with full details on the experimental settings and hyperparameter choices.
>
> [1]New insights and perspectives on the natural gradient method, Martens, 2020
> [2] BackPACK, Dangel et al., 2020
> [3] Liberty or Depth: Deep Bayesian Neural Nets Do Not Need Complex Weight Posterior Approximations, Farquhar et al. 2021

---

> ### Author Response · Authors · 2024-04-10
> **Rebuttal pt. 2**
>
> 4. The penalty arises from the interplay between the expectation of the loss w.r.t. the posterior and the KL divergence which penalizes departures from the prior distribution.  Larger penalty values force posterior variational distributions with larger volume, which increases penalization due to the expectation of the loss over the posterior.  It is this expectation term that is key.  To see how penalization works in a simplified setting, we can ignore the prior term and, choosing $q \sim \mathcal{N}(w, \Sigma)$, this leads to the following variational objective:
> \begin{equation}
> \arg \min_{q}\mathbb{E}_{q}[L(w)]  - \rho \mathcal{H}(q) = \arg \min_q \mathbb{E}_q[L(w)]  + \rho \log \frac{1}{\det(\Sigma)}
> \end{equation}
> so that large values of $\rho$ will correspond to distributions with larger volume, since for $x \sim \mathcal{N}(0, \Sigma)$, $x$ lies within the ellipsoid $x^T {\Sigma}^{-1} x =  \chi^2(\alpha)$ with probability $1-\alpha$, with the volume of the ellipsoid proportional to $\det(\Sigma)^{\frac{1}{2}}$.
>
> More generally, we show in the updated manuscript how the G-TRACER penalty term $\rho \text{Tr}(G^{-1} F_w)$ corresponds to integrating over the posterior, corresponding to a kernel smoothing of the loss-surface with a Guassian kernel, and also provide an interpretation of the penalty as a metric trace of the Hessian (or Laplace-Beltrami operator) which measures the difference between the mean value of a function over a geodesic ball and the value at a minimum.  This Hessian metric trace, which adjusts for the local geometry of the loss, also appears in a general form (for a general metric) as a potential sharpness measure in a recent NeurIPS paper [4].
>
> 5. Thank you for raising this interesting point.  Input/label noise can be transferred to the weights.  Consider the single-layer case with weight matrix $W$ and input $x$: we have $L(W + \Delta W,x) = L(W,x+\Delta x)$ for the choice $\Delta W = \frac{W \Delta x} {||x||_2^2} x^T$  (indeed, there are infinitely many solutions to the underlying matrix equation) so that input noise robustness corresponds to robustness in weight-space.  This type of construction can be generalized to deeper architectures (see eg [5],[6]).  Methods like G-Tracer and SAM which are robust to weight perturbations are for this reason expected to be robust to noise.  Indeed, the most convincing and striking results in the original SAM paper concern robustness to label noise.
> 6. a. & b. This is a very helpful observation and is now reflected in the updated manuscript.  This particular construction (which we qualified in the text as "informal") was intended only to provide motivation and context and indeed only applies to the case where minima are isolated.  To avoid confusion and to shorten the paper, we have removed this section entirely.
>    c. At a critical point, the Hessian eigenvalues are the principal curvatures.  Scaling the Hessian by $\epsilon$ scales the eigenvalues of the Hessian by $\epsilon$, and thus the principal curvatures.<br>
> 7. Noted and changed.
>
> [4] The Geometry of Neural Nets' Parameter Spaces Under Reparametrization, Kristiadi et al., 2023
> [5] Towards Flatter Loss Surface via Nonmonotonic Learning Rate Scheduling, Seong et al., 2018
> [6] Generalization in fully-connected neural networks for time series forecasting, Borovykh et al., 2019

---

### Review · Reviewer_NxpK · 2024-01-22

**Summary Of Contributions:**

The paper proposes a new algorithm align the line of sharpness aware minimization, motivated from a statistical heavy perspective on minimizing curvature in expectation. Some experiments are shown to highlight better generalization of the proposed algorithm.

**Audience:**

Yes

**Broader Impact Concerns:**

No concerns.

**Claims And Evidence:**

No

**Requested Changes:**

Please address the experiments weakness I mentioned, the current experiments does not seem really trustworthy given the parameter choices and seemingly low accuracies across the board. Also adding comparison with one or two state-of-the-art improvements over SAM can strengthen the paper in my perspective, both in literature review and experiments.

**Strengths And Weaknesses:**

Strengths:
1. The paper is well-written and well-organized, with a review on motivations and explanations of sharpness aware minimization, highlight weakness of the original sharpness aware minimization (SAM). Then the proposed algorithm is motivated and explained step-by-step.
2. The proposed algorithm is backed with theories to show how intuitively it can outperform SAM and the algorithm is proven to converge to a neighborhood of stationary points of original problem.

Weaknesses:
1. The major weakness is experiment evaluation of the proposed algorithm. All algorithms are compared using a single learning rate, instead of tuning and find the best learning rate for each algorithm, the learning rate seems to be decided in an adhoc way, it is hard to evaluate the fairness of the comparisons. Also, reported accuracies for CIFAR-100 with resnet-18 seems low, it should achieve well-above 70% accuracy without label noise using open-source code online, however the paper only reported around 70% accuracy, the models may not be well-trained.
2. There is no comparison with other variants of SAM, also there is not sufficient literature review on recent improvements to SAM.

Overall the paper has nice theories and stories that could be of interest to some audience, but the proposed algorithm is not backed by convincing experiments.

---

> ### Author Response · Authors · 2024-04-10
> **Rebuttal**
>
> We thank the reviewer for the very helpful review.  We fully agree that the experimental setup was not clearly set-out and have fully elaborated upon this in the updated version, in which we have significantly strengthened the results section and added further experimental results.
>
> Taking each point in turn:
>
> 1. The exact experimental setting for the vision tasks (unless otherwise indicated) follows standard practice and is as follows: SGD with weight decay/ridge penalty $5 \times 10^{-4}$, momentum $0.9$, initial learning rate $0.1$, 200 episodes, cosine learning rate decay to $0$, batch size 128, global clip-norm$=1.0$.  The search spaces for the SAM and G-Tracer penalties $\rho$ were chosen by first running on a logarithmic grid of 10 values $[1 \times 10^{-5},\dotsc,1 \times 10^4]$ and then refining the range based on in-sample convergence, in order the span the space of plausible regularization strengths.  These results for ResNet-20 are in line with (in fact, competitive with) the results in [1] and [2] .  As a further consistency check with practice, follow the training protocol (stepwise learning rate decay over 200 episodes, with learning rates $[.1, .02, .004,.0008]$ at $[0, 60,120,160]$) in https://github.com/weiaicunzai/pytorch-cifar100/tree/master?tab=readme-ov-file with larger architectures, eg ResNet-18 (11M parameters), and match the expected results for SGD, and see similar improvements vs SGD (75.8\% accuracy vs 75.1\% accuracy) and SAM (75.3\% accuracy, $\rho=.05$).
> 2. We have added an extensive literature review and benchmarked our baseline results against those in the key papers [1] and [2] so tha comparisons can be made against other variants.
>
> More generally, while the original SAM paper (and related literature) largely focus on standard benchmark problems and show marginal improvements in many settings, the most striking and practically relevant improvements concern the performance gains in the more challenging noisy-label settings.  The CIFAR-10/100 benchmarks are extremely well understood, and good training schedules, data augmentations, and architecture choices have all been found over an extremely large number of trials run by the community over many years.  The effect size of augmentations is often large (10\% accuracy gains, or more) compared to post-augmentation gains exhibited by SAM (typically on the order of 1\%).  In contrast, our particular focus is on practically relevant regularization that is expected to boost performance in challenging low signal-noise ratio problems, in the absence of data augmentations and other regularizations, and our "model" for this setting is challenging variants of known benchmarks, eg training vision tranformers (ViT) from scatch (no pretraining), heavily noise currupted CIFAR-100 with no augmentation, as well as 3 challenging NLP tasks in Bert fine-tuning (BoolQ, WIC, RTE).
>
> [1] Thomas Möllenhoff and Mohammad Emtiyaz Khan. Sam as an optimal relaxation of Bayes, 2023.
> [2] Jungmin Kwon, Jet. al, Adaptive sharpness-aware minimization for scale-invariant learning of deep neural networks, 2021

---

### Review · Reviewer_QV3t · 2024-01-29

**Summary Of Contributions:**

The paper introduces TRACER as a regularizer to first-order methods, such as SGD and Adam. The resulting algorithm favors flat local minima and thus improving the generalization of the trained models. The improved performance is demonstrated by experiments. Major contribution of the paper is the proposal of TRACER.

**Audience:**

Yes

**Claims And Evidence:**

No

**Requested Changes:**

Besides the presentation, I have another comment on the experiment section. From the pure classification accuracy, we find SGD-TRACER yields the best performance. However, how does the performance related to the flat minima? There should be some exposure of the flatness of the minimizer that is found by SGD-TRACER. This is an important verification of the proposed algorithm.

**Strengths And Weaknesses:**

Although I believe that TRACER is a new idea in the literature and the experimental performances demonstrate its strength, it is relatively difficult to appreciate the novelty in the current presentation:

1. The paper starts with problem setting in the introduction section. I would rather put the content in a preliminary section. There are quite a few mathematical notations to digest.

2. The paper indeed lacks an introduction section. There is no explicit motivation of the study, and no summary of the contributions as well as its relevant literature.

3. The presentation in the paper is rather casual. For example, there are unsupported claims, e.g., "There has therefore been a large literature attempting to characterize the loss-surface Hessian $\nabla^2 L(w)$ and to relate these characteristics to generalization." without any references after equation (8).

There are unjustified approximate equalities, e.g., "Equation (2)" without any details on the Laplace transform and what is hidden in the approximate equal sign.

There are also quite a few typos and grammatical issues, e.g., "Equation (1) ends without any period"; "double 'can' in the sentence following Equation (1)".

I would encourage the authors to proofread the paper thoroughly.

---

> ### Author Response · Authors · 2024-04-10
> **Rebuttal**
>
> We are extremely grateful for the reviewer's comments.
>
> Taking each of the points in turn:
>
> 1 & 2.  We fully agree.  We have fully reworked the exposition to clarity the material and aid understanding, and have added derivation sketches and moved the detailed derivations to appendices.
> 3. Full agreed - we have reworked the paper extensively with this comment (and others like it) in mind.
>
> The Laplace transform is not used anywhere in the paper; however the notation and exposition has been significantly improved.
>
> Regarding requested changes, we fully agree and have set out in detail the theory linking geometric flatness and the G-Tracer regularization scheme, in parrticular, highlighting its connection to the Laplace-Belrami Operator and, in particular how, at a critical point, $\mathrm{Tr}({G}^{-1}H)$ is the Laplace-Beltrami operator $\Delta$ (also known as the manifold Laplacian) which generalizes the Laplacian to Riemannian manifolds [1], and defines an invariant, geometric quantity which, by analogy with $\text{Tr}(H)$ in Euclidean space, measures the average deviation from flatness, adjusting for the curvature of the manifold.  Crucially, this notion is not an assumption, but rather emerges naturally from the variational optimization of a generalized Bayes objective using the KL-metric.  In particular, for a multivariate Gaussian variational approximation, the trace penalty corresponds to a smoothing of the loss surface using a kernel estimated online.
>
> In more detail, in the neighborhood of a local minimum ${\mu^*}$ of $L$, the Laplacian can be interpreted as the difference between the mean value of $L$ over a (geodesic) ball centered at ${\mu^*}$ and $L({\mu^*})$, due to the following mean-value property for smooth functions over geodesic balls $B_r({\mu})$ [2]:
> \begin{equation}
>     \frac{1}{\mathrm{vol}(B_r({\mu^*}))}\int_{B_r({\mu^*})}L({\mu}){dV} - L({\mu^*}) = \frac{{\Delta} L({\mu^*})}{2(n+2)}r^2 + \mathcal{O}(r^4)
> \end{equation}
>
> [1] John M. Lee, Introduction To Riemannian Manifolds, 2018
> [2] A. Gray and T. J. Willmore. Mean-value theorems for riemannian manifolds. Proceedings of the Royal
> Society of Edinburgh: Section A Mathematics, 92(3–4):343–364, 1982.

---

### Decision · Action_Editor_NHcA · 2024-04-06

**Recommendation:** Reject

**Comment:**

There are a few major concerns on the paper:
1) The paper only provides experimental evaluation of the proposed algorithm on two relatively simple tasks.
2) The method requires computation of the Hessian at each step, as well as an inverse of Hessian which is even more computationally costly, making it less scalable than SAM.
3) The paper is written in an somewhat unusual way, starting from the formulation in the first page, without introducing background and summarizing contributions first.

In addition, there are no responses to the reviews.

**Audience:**

Yes. If the findings are fully justified, then the findings will be of interest to neural network researchers.

**Claims And Evidence:**

No. The paper only provides experimental evaluation of the proposed algorithm on two relatively simple tasks, which is not convincing.